# Topological analysis of multicellular complexity in the plant hypocotyl

**Matthew DB Jackson, Hao Xu, Salva Duran-Nebreda, Petra Stamm, George W Bassel\***

School of Biosciences, University of Birmingham, Birmingham, United Kingdom

**Abstract** Multicellularity arose as a result of adaptive advantages conferred to complex cellular assemblies. The arrangement of cells within organs endows higher-order functionality through a structure-function relationship, though the organizational properties of these multicellular configurations remain poorly understood. We investigated the topological properties of complex organ architecture by digitally capturing global cellular interactions in the plant embryonic stem (hypocotyl), and analyzing these using quantitative network analysis. This revealed the presence of coherent conduits of reduced path length across epidermal atrichoblast cell files. The preferential movement of small molecules along this cell type was demonstrated using fluorescence transport assays. Both robustness and plasticity in this higher order property of atrichoblast patterning was observed across diverse genetic backgrounds, and the analysis of genetic patterning mutants identified the contribution of gene activity towards their construction. This topological analysis of multicellular structural organization reveals higher order functions for patterning and principles of complex organ construction.

**\*For correspondence:** g.w. bassel@bham.ac.uk

**Competing interests:** The authors declare that no competing interests exist.

## Introduction

Multicellularity arose multiple times across evolution (*Kaiser, 2001*; *Knoll, 2011*), yet how selective pressures shaped and optimized the cellular configurations of these complex assemblies remains poorly understood (*Ollé-Vila et al., 2016*). Multicellular organs are more than the sum of their cells, and the collective interactions between cells on a global scale confer higher order functionality to the system through a structure-function relationship (*Thompson, 1942*). Cellular functionality therefore emerges from cellular associations and synergies, and is not cell autonomous. Understanding the emergent properties of complex multicellular assemblies, and the structure-function relationship between cell organization and organ function, remains an open challenge in both developmental and systems biology.

This question has been examined previously in the field of neuroscience in the investigation of the relationship between cellular organization and nervous system function (*Cajal, 1911*; *White et al., 1986*). This was first systematically applied to the simple nervous system of *C. elegans* (*White et al., 1986*), and more recently the field of 'connectomics' has extended this approach to more complex nervous systems (*Bullmore and Sporns, 2009*). Here a distinction between structural and functional networks is drawn, the former being the physical associations between cells representing all possible routes of information flow, and the latter the paths which information is observed to follow (*Bullmore and Sporns, 2009*).

Uncovering the organizational properties of complex multicellular assemblies has not been performed previously at a whole organ or organism level. In plants, cells are glued together through shared cell walls and do not migrate with respect to one another, as in animal systems (*Green, 1969*). This invariance between adjacent cells provides a simplified opportunity to examine multicellular complexity by looking at whole organ cellular interaction networks that remain topologically

**eLife digest** In plants and other multicellular organisms, cells work together in tissues and organs to achieve outcomes that they would not be able to accomplish on their own. For example, the cells that make up the stem of a plant hold the leaves, flowers and other organs in place, and provide a transport network that allows molecules to move around the plant. Mapping the locations of cells within tissues and analysing the connections between them will help researchers to understand how the organisation of tissues influences the tasks cells perform.

There are several different layers of tissue within a plant's stem. The surface of the stem has a protective layer of tissue called epidermis. The epidermis contains two different types of cells known as trichoblasts and atrichoblasts, but it was not clear why these cells are organised the way they are.

*Arabidopsis thaliana* is a small plant that is often used in studies of how plants grow and develop. Jackson et al. combined microscopy with computational techniques to study the stems of young *A. thaliana* seedlings. The experiments reveal that the two types of epidermal cells appear to adopt distinct roles. The trichoblasts form hair-like structures and acquire nutrients from the external environment, while their neighbours the atrichoblasts provide shortcut routes for these nutrients to be unloaded and moved up the stem. This pattern was not present in several other plant species including foxglove or poppy, suggesting it may be an adaptation in *A. thaliana* plants that helps them grow in the particular environments this plant faces.

The findings of Jackson et al. show that cells are carefully arranged in plant stems and suggest that there is an optimal way for a plant to make a stem depending on its environment. Further work is now needed to understand how different molecules use the shortcuts provided by the atrichoblasts during plant development, and whether alternative configurations are possible. In the future, such studies may help provide a framework to genetically engineer plants that are better adapted to grow in different environments.

invariant following their formation. By viewing plant organs as a complex system of interacting cells, a systems-based approach to understanding organ construction and optimization at a cellular level can be undertaken.

Cellular interactions play a key role during plant development (*Benitez-Alfonso et al., 2013*; *Lucas and Lee, 2004*). Mobile information in the form of proteins, RNAs and small molecules move locally through physical cell-to-cell interactions. These local interactions mediate patterning, self-organization and underlie cell identity in plants (*Leyser, 2011*; *Sabatini et al., 1999*; *Sena et al., 2009*; *Sugimoto et al., 2010*). Genetically encoded patterning mechanisms mediate the self-organization process that leads to the creation of functional cellular interactions and patterns that constitute plant organs (*Besson and Dumais, 2011*; *Di Laurenzio et al., 1996*; *Yoshida et al., 2014*).

While the importance of intercellular interactions is well established, much less is known about the global properties of these assemblies, and how they come together to form coherent organs. Previous efforts to understand local interactions between cells in two-dimensional cellular sheets have been explored using the developing *Drosophila* wing disk. Here a network consisting of nodes representing cell-cell junctions was generated, and cell shape based on polygon classes (or local connectivity) was generated (*Gibson et al., 2006*, *2011*; *Heller et al., 2016*). In plants, local cellular interactions and their role in mediating cell division plane placement in neighbors has also been investigated (*Gibson et al., 2011*; *Sahlin and Jönsson, 2010*; *Willis et al., 2016*). While these approaches are informative, they are limited to the local topological analysis within the immediate vicinity of a cell, and are also restricted to cells on the surface of organs.

Advances in whole organ 3D imaging at cellular resolution and computational image analysis enable organ-wide cellular interaction networks to be extracted and annotated by cell type (*de Reuille et al., 2015*; *Montenegro-Johnson et al., 2015*) (*Figure 1A*). This multidimensional topological phenotyping pipeline captures global cellular interactions in whole organs and enables the simultaneous analysis of both higher-order and local properties of these complex cellular configurations. These structural networks provide cellular roadmaps of possible information flux through

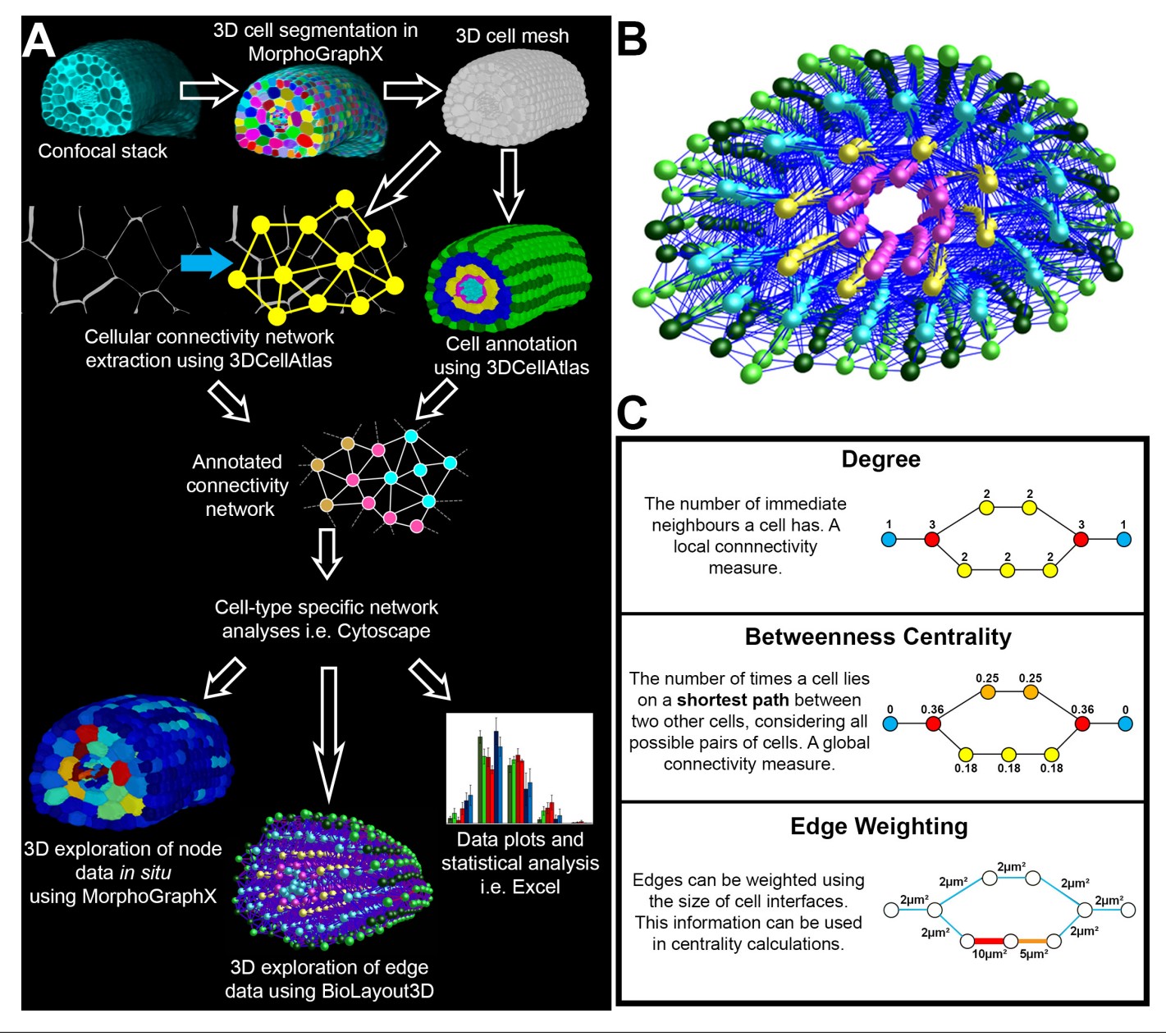

**Figure 1.** Computational workflow. (**A**) Pipeline for imaging, annotating, quantitatively analyzing, and visualizing global cellular interaction networks. (**B**) 3D layout of an annotated cellular connectivity network for the Arabidopsis Colombia hypocotyl. (**C**) Explanation of the network measures employed in this study. Nodes are colored to indicate values.

The following figure supplements are available for figure 1:

**Figure supplement 1.** Filtering small edges in connectivity networks.

**Figure supplement 2.** Topological buffering of sample boundaries.

organs, and understanding the structure of these cellular communities provides insight into the function of the system as a whole, and individual cells within organs.

We examined the embryonic plant stem (hypocotyl) to understand the principles with which cells come together to create this organ. This radially symmetric multicellular assembly elongates

exclusively through cell expansion during early seedling establishment, rendering cellular topology invariant across development. The hypocotyl also serves as a conduit linking the above and below ground portions of the seedling during early growth, serving both a transport and structural function.

## Results

### Extraction and analysis of cellular connectivity networks

In order to understand both the local and higher-order properties of complex multicellular organization in whole organs, an image analysis and analytical framework was employed (*Figure 1A*). To achieve robust topological analyses of cellular patterning in plant organs, we sought to generate fully represented and accurate cellular interaction networks. Every cell within the unexpanded embryonic hypocotyl was digitally captured using whole mount high resolution confocal microscopy and 3D segmentation (*de Reuille et al., 2015*; *Montenegro-Johnson et al., 2015*) (*Figure 1A*). Cell types within these samples were identified and annotated using 3DCellAtlas (*Montenegro-Johnson et al., 2015*). Cellular connectivity networks were extracted by identifying shared surfaces between adjacent cells using 3D cellular meshes, representing intercellular associations. Organ-wide cellular connectivity networks that are annotated by cell type were generated (*Figure 1A–B*, *Video 1*). Towards the achievement of accurate cellular interactomes, segments representing air spaces between cells were computationally removed (*Montenegro-Johnson et al., 2015*). Relatively small interfaces (below 2 μm$^2$) were filtered out due to their likely limited role in transport processes, and potential generation due to image processing artefacts (*Figure 1—figure supplement 1*) (*Bassel, 2015*). Cells in the periphery of imaged organs were included in analyses, but only cells from the central regions of organs were reported in figures (*Figure 1—figure supplement 2*). This provided a topological buffer to the boundaries introduced following image acquisition. Collectively, these processes yielded precise global cellular interaction networks annotated by cell type, enabling cell type specific topological analyses of organ patterning to be performed.

A persistent issue with analyses using biological network datasets is their completeness and accuracy. The networks presented in this work are both fully represented (no missing nodes or edges) and highly accurate. In all instances, biological triplicates are used for the analysis of organ-wide patterning in this study.

A remaining challenge is to identify an appropriate analytical framework to extract biologically meaningful information from these connectivity datasets. Considering the importance of intercellular interactions within plant organ function, these networks may be viewed as a complex system (*Newman, 2010*). The spatial constraints of plant cells and their embedding in space mean these networks resemble 3D lattices (*Barthelemy, 2011*). The properties of these undirected cell interaction networks can therefore be analyzed using quantitative network analyses to reveal the organizational properties of these multicellular assemblies (*Barabási, 2016*; *Bullmore and Sporns, 2009*). Following topological analyses, node data can be visualized by importing cell properties back into MorphoGraphX (*de Reuille et al., 2015*) as a heatmap for in situ 3D visualization within the cellular context of the segmented organ (*Figure 1A*). Edge information can be graphically explored by importing these data into the BioLayout3D visualization tool and using the edge heatmap functions (*Figure 1B*) (*Theocharidis et al., 2009*). Finally, statistical analyses of data exported from network analysis software can be performed using standard statistical software (*Hagberg et al., 2008*).

This approach enables the capture, discretization and quantification of local and global cellular organization, or patterning, in whole organs.

A simple measure of the local properties of a cell (node) within an organ (network) is to count

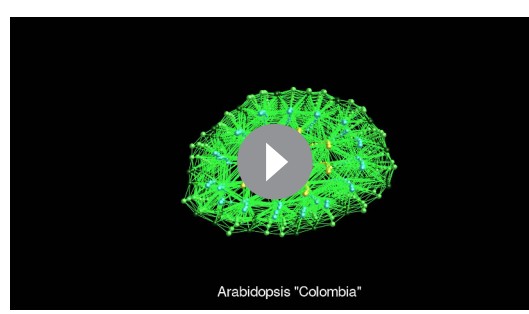

**Video 1.** Hypocotyl cell connectivity network of *Arabidopsis thaliana* Col. Cell types are gradually removed to allow internal visualisation of the network.

the number of neighbours it has (*Figure 1C*). This is termed the degree of a node, and is a local measure of how a cell fits within its immediate context (*Newman, 2010*). While informative, degree is unable to capture the higher-order properties of multicellular systems (*Gibson et al., 2006*, *2011*; *Heller et al., 2016*; *Newman, 2010*; *Willis et al., 2016*).

In light of the 3D spatial constraints of immobilized plant cells within organs, and the central role of intracellular communication in plant development, path length represents an important and biologically relevant topological property of these networks (*Barthelemy, 2011*). A well-established measure of path length in networks is betweenness centrality (BC) (*Barabási, 2016*; *Freeman, 1977*; *Newman, 2010*) (*Figure 1C*). BC uses knowledge of the complete network to count the number of times a nodes lies on a shortest path between all other nodes (*Barabási, 2016*; *Newman, 2010*). A node with a high BC therefore lies on many shortest paths between other nodes. In this study we normalized BC by network size in order to make data comparable between different samples (*van Wijk et al., 2010*).

The use of BC enables the identification of short paths through multicellular networks, representing topologically poised optimized routes for information movement across the system. This combination of acquiring complete cellular interactomes and their analysis using path-length-based measures enables the higher-order principles of global cellular patterning to be revealed.

## Topological analysis of the wild-type Colombia *Arabidopsis* hypocotyl

The properties of three wild-type *Arabidopsis* hypocotyls of the ecotype Colombia (Col) were topologically analyzed at cell type specific resolution (*Figure 2A*). To illustrate key findings, we focus the reporting of results on epidermal cell patterning, with additional cell type analyses presented as corresponding figure supplements.

The spatial distribution of cell (node) topological properties can be visualized by false coloring cells of segmented organs using centrality data (*Figure 2B–C*) (*de Reuille et al., 2015*). This provides an in situ view of the distribution of node properties across the segmented organ within individual cells. Cell degree (number of immediate neighbors) is greater in cortical cells than in the epidermis (*Figure 2B* and *Figure 2—figure supplement 1*) as this latter cell type lacks exterior neighbors. Degree is also high in elongated vascular cells representing xylem vessels (*Figure 2B*).

Node BC is low in most cells of the *Arabidopsis* Col hypocotyl, and cells lying upon shorter paths (higher BC) are observed in a small fraction of the network, as observed by the presence of a tail in this distribution (*Figure 2E* and *Figure 2—figure supplement 1*). The highest node BC, representing the shortest paths across the organ, is observed in the vascular system (*Figure 1C*), consistent with the role of this cell type in facilitating long distance transport.

High BC nodes are also observed in epidermal cell files belonging to the non-hair forming atrichoblast cell type (*Duckett et al., 1994*) (*Figure 2C and E*). These cells lie on significantly shorter paths than their adjacent cell type, the hair forming trichoblast cells (*Figure 2E*). This topological analysis identified a previously undescribed coherent conduit of interconnected cells providing a short path length across the atrichoblast cell type of the *Arabidopsis* Col hypocotyl epidermis (*Figure 2C and E*). These data demonstrate that in addition to the vasculature, atrichoblast cell files are topologically poised to mediate the optimized movement of information along the longitudinal length of the hypocotyl axis.

While the degree of trichoblast cells is significantly greater than that of atrichoblast cells (*Figure 2D*), it is these latter cells with fewer connections that lie upon shorter paths. This represents a non-intuitive global topological property of this multicellular system. The higher order properties of cells within the hypocotyl are therefore not dictated by their number of local interactions, but rather their context within the wider system.

We next examined each the geometric and topological properties of the interfaces, as opposed to the nodes, between adjacent cells. These are represented as edges within cellular connectivity networks. These intracellular interfaces provide the physical conduits by which information moves through these multicellular systems (*Figure 2F* and *Video 1*).

The geometric size of these shared interfaces between cells was computationally measured using 3DCellAtlas (*de Reuille et al., 2015*; *Montenegro-Johnson et al., 2015*), and examined across the unexpanded embryonic Col hypocotyl (*Figure 2—figure supplement 2*). In absolute terms, cellular interactions between the large cortical cells had the greatest interface sizes, reflecting differences in

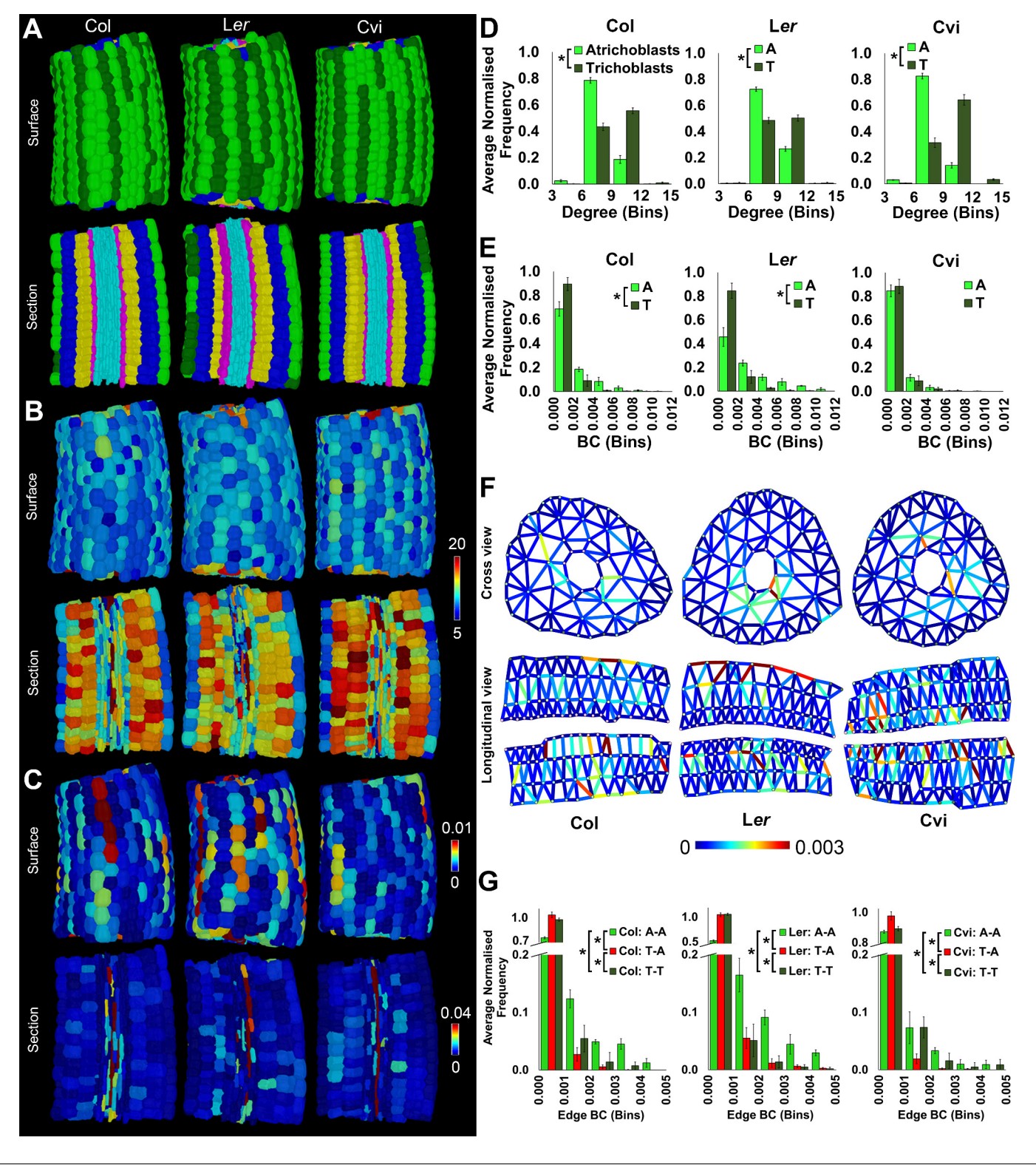

**Figure 2.** Cell type-specific quantification of the topological features of hypocotyl interaction networks in three Arabidopsis ecotypes: Colombia (Col), Landsberg erecta (Ler) and Cape Verdi Islands (Cvi). (**A**) Surface and longitudinal-section meshes of the three ecotypes, with false color indicating cell type (dark green – trichoblast, light green – atrichoblast, blue – outer cortex, yellow – inner cortex, pink – endodermis, cyan – vasculature). (**B–C**) Hypocotyl meshes with false color heat maps of (**B**) degree and (**C**) betweenness centrality. (**D–E**) Average normalized frequency distributions of (**D**) degree and (**E**) BC across trichoblast and atrichoblast cell types. (**F**) Edge BC false colored onto edges of virtual cross and longitudinal sections of the

*Figure 2 continued on next page*

*Figure 2 continued*

hypocotyl networks of each ecotype. (**G**) Normalized frequency distributions of edge BC in different cell interfaces within different cell types (T – trichoblast, A – atrichoblast). (**H**) Schematic showing cells lying upon short path lengths within the *Arabidopsis* hypocotyl. Biological replicates were treated as individual samples and data represent the mean frequency of the bins across the triplicate samples. Error bars represent ± standard deviation within a frequency bin. Asterisks (*) indicate significant differences between distributions using the chi-squared test for degree and the Kolmogorov–Smirnov test for BC, at the $p \leq 1.56 \times 10^{-5}$ level ($p \leq 0.05$ after Bonferroni correction for 3200 distribution comparisons).

The following figure supplements are available for figure 2:

**Figure supplement 1.** Comparisons of topological features of *Arabidopsis* hypocotyl cellular arrangements between Arabidopsis ecotypes.

**Figure supplement 2.** Cell interface sizes in unexpanded embryo hypocotyls.

**Figure supplement 3.** Cell surface area and weighted BC measurements.

cell size, and cells smaller in size had reduced contact areas with their neighbors (*Figure 2—figure supplement 2*).

Topological analysis of these cell interfaces is provided using edge BC, which measures the participation of edges in the creation of short paths between all pairs of nodes (*Newman, 2010*). Atrichoblast-atrichoblast edge BC was greatest of all cellular junctions between different cell types within the epidermis (*Figure 2G* and *Video 2*). This demonstrates that intercellular interactions between atrichoblast cells contribute most towards the creation of short path lengths across this cell type.

Both cellular connectivity networks and cell interaction sizes can be extracted from whole organs using 3DCellAtlas. This enables the integration of these two pieces of information by weighting edges in the network with the geometric size of their associated cellular interfaces. In this case, weighted node betweenness centrality was calculated using the reciprocal of the connecting cell wall area (edge weight), so that large cell-cell interactions generate small weights, which contributes more to the creation of short paths (*Newman, 2010*). The topological analysis of weighted hypocotyl networks generated a bias based on cell size whereby weighted BC identified large cortical cells as having the lowest path length (*Figure 2—figure supplement 3*). The superior size of cortical cells in the hypocotyl endow these cells with a geometrically facilitated reduction in path length in these weighted networks. Information flow which is limited by the size of cellular interfaces may therefore prefer to move through this cell type, however considering the scale of cellular interface sizes included in networks which is in the order of microns (*Figure 2—figure supplement 2*), this does not likely represent a limiting factor in this context.

Increased surface area has biological relevance to mediating intercellular trafficking through molecular transport components, which occupy physical space. The role of cell interface size towards mediating topologically distinguishing features however appears to be limited in the context of the unexpanded embryonic hypocotyl in *Arabidopsis* Col due to large differences in size across cell types.

## Comparison of topological features across *Arabidopsis* ecotypes

Wild *Arabidopsis* plants have been collected from diverse locations across the planet, facilitating the study of naturally occurring variation within this species (*Koornneef et al., 2004*). Cellular organization in the Cape Verdi Islands (Cvi) and Landsberg *erecta* (L*er*) ecotypes was compared with that of Col to explore topological

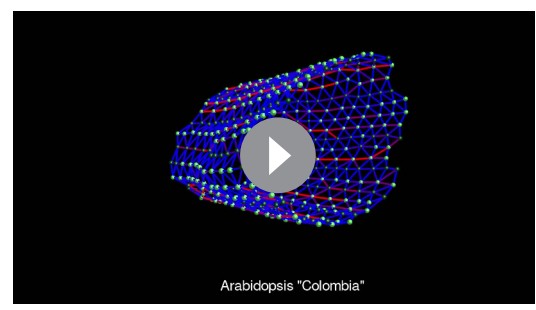

**Video 2.** Hypocotyl cell connectivity network of the epidermis of *Arabidopsis thaliana* Col. Edge colour represents the edge random walk betweenness centrality (scale 0–0.0015, blue to red).

principles of patterning across diverse genetic backgrounds.

Local epidermal patterning was similar in all ecotypes, with the trichoblasts (hair forming cells) having a significantly higher local connectivity than the atrichoblast cells (*Figure 2D*). The degree of Cvi trichoblast cells was greater than the other ecotypes, and the atrichoblast cells of L*er* also had a greater number of neighbors (*Figure 2D*, *Figure 2—figure supplement 1*).

The distributions of node BC were similar in the trichoblast cells of all three ecotypes, however path length distributions varied in atrichoblast cells across these different genetic backgrounds (*Figure 2—figure supplement 1*). Comparing these two cell types within each ecotype, significantly atrichoblast cells lay upon significantly shorter paths than trichoblasts in Col and L*er*, while no significant difference was present in Cvi (*Figure 2E*). These observations indicate the presence of natural variation in path length, specifically within the atrichoblast cell type of the *Arabidopsis* hypocotyl. This topological plasticity in turn leads to contrasting path lengths between each of the two epidermal cell types in Col and L*er*, but not the Cvi accession.

## Quantification of cell type-specific molecular movement through *Arabidopsis* hypocotyls

The optimized movement of information through networks follows shortest paths (*Barabási, 2016*; *Newman, 2010*). Topological analyses of *Arabidopsis* hypocotyl cellular interaction networks revealed the presence of reduced path lengths in the epidermis specifically within the atrichoblast cell files of the Col and L*er* ecotypes, but not that of Cvi (*Figures 2E* and *3A*). We examined these topological predictions of preferential cell type-specific flux of molecular information by tracking the movement of the small fluorescent molecule fluorescein. Seedlings were transferred to media containing fluorescein, and following an incubation period, hypocotyls were live imaged using confocal microscopy. Epidermal cells in the hypocotyl typically do not produce root hairs, despite having each trichoblast and atrichoblast cell identities. The observation of the passage of fluorescein along the hypocotyl occurs following molecular uptake in roots, where hairs are produced, and represents long distance transport along the length of the plant rather than local uptake.

Z-stacks were obtained and epidermal cells were segmented in 3D to quantify the concentration of fluorescein within each epidermal cell type (*de Reuille et al., 2015*). This enabled the transport of this molecule through these multicellular networks to be captured at single cell resolution. Considering plants do not produce fluorescein, no specific transporter for this molecule is likely to be present (*Barclay et al., 1982*). These measurements may therefore capture the non-specific bulk flow of molecules from the site of uptake in the roots, up through the hypocotyl.

In all ecotypes, fluorescein was observed to be transported along the hypocotyl of seedlings examined (*Figure 3B–D*). In each L*er* and Col, a qualitatively greater amount of fluorescein was observed in atrichoblast than trichoblast cells (*Figure 3B–C*), and this distinction was less clear in Cvi (*Figure 3D*). Quantification of fluorescein concentration within each of these cell types across ecotypes (*Figure 3E–G*) revealed a significantly greater abundance of this molecule in atrichoblast cells over trichoblast cells in L*er* and Col, and no significant difference between the cell types was observed in Cvi (*Figure 3H*).

These measurements support the link between path length analyses and the bulk movement of molecules along the epidermis in hypocotyls. The reduced path length in the atrichoblast cells of L*er* and Col predict the preferential movement of small molecules through this epidermal cell type, while in Cvi where path length is equivalent between both epidermal cell types, no bias in molecular movement is observed. This suggests that BC can be used to predict the long range movement of molecules in this multicellular plant organ at single cell resolution. The construction of these conduits through the emergent property of global patterning therefore plays a functional role in optimized molecular transport across this organ. In the absence of active transport processes, molecules will preferentially move following these paths of reduced path length. Components participating in the active transport of molecules may also exploit these optimized routes by populating these cells with transport machinery.

This higher order property of multicellular organization may also have a functional role within the context of epidermal cell patterning (*Dolan et al., 1993*). The acquisition of nutrients in the soil by root hairs (trichoblasts) is dependent upon the ability to import these into cells. The accumulation of imported solutes within a root hair would impede this process owing to increased concentration within a cell as compared with that in the soil. A solution to this could be to acquire nutrients in root

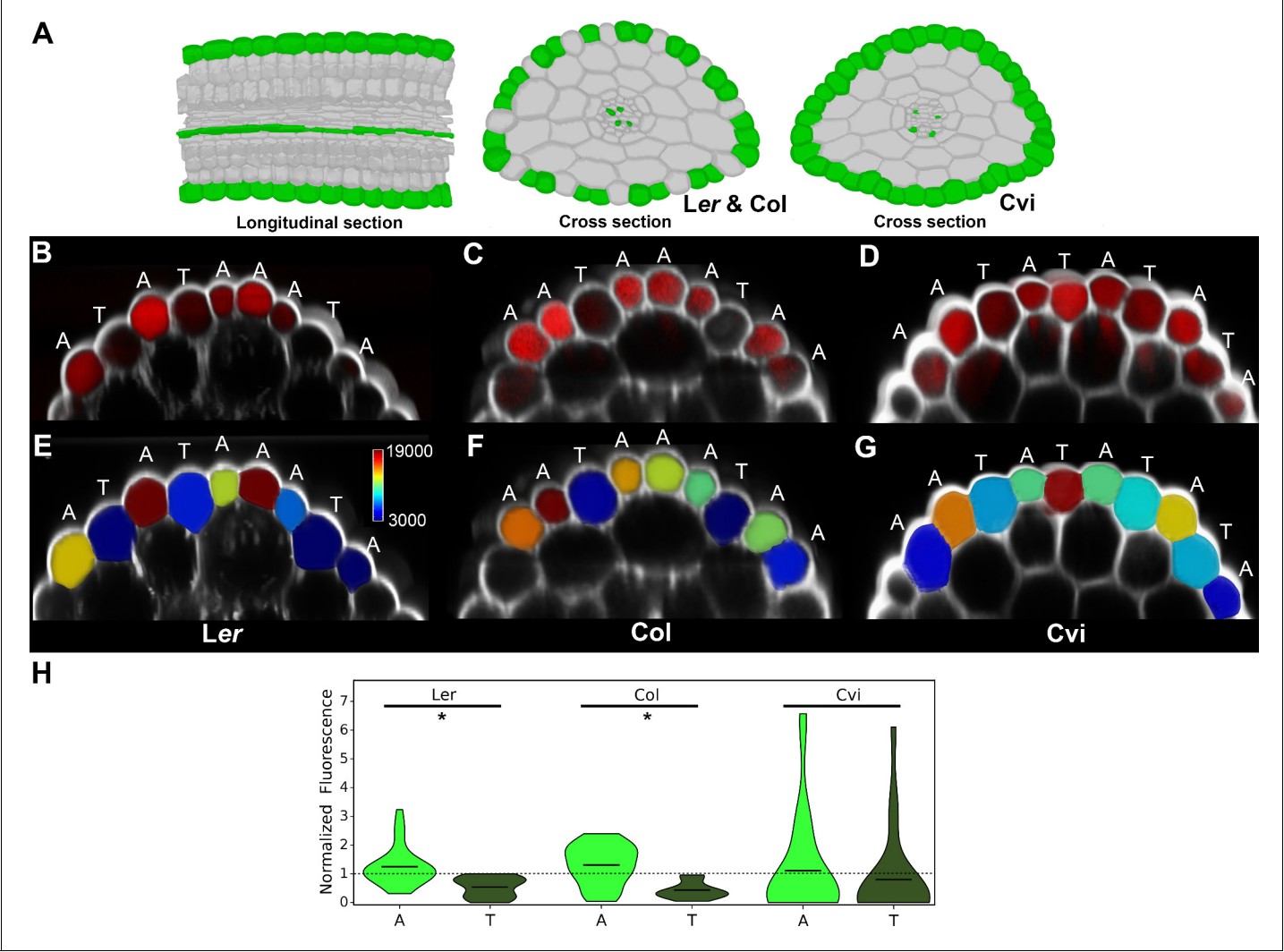

**Figure 3.** Transport of the small fluorescent molecule fluorescein in the hypocotyl epidermis of different *Arabidopsis* ecotypes. (**A**) Illustration of the cells having reduced path length in the hypocotyls of Landsberg *erecta* (L*er*), Colombia (Col) and Cape Verdi Islands (Cvi) ecotypes indicated in green. (**B**) Confocal image of the L*er* hypocotyl (white) and fluorescein (red) imaged within the epidermis. (**C**) Same as (**B**) with Col. (**D**) Same and (**B**) with Cvi. (**E**) Quantification of fluorescein concentration within individual epidermal cells of the L*er* hypocotyl epidermis. (**F**) Same as (**E**) with Col. (**G**) Same as (**E**) with Cvi. Atrichoblast and trichoblast cell types are indicated by an A or T above each cell. The scale bar in (**E**) indicates mean normalized values for the quantification of fluorescein concentration visualization in (**E**)-(**G**). (**H**) Violin plot of fluorescein concentration in each atrichoblast and trichoblast cells. The mean is indicated by a black bar. An asterisk (*) indicates a significant difference in fluorescein concentration between atrichoblasts and trichoblasts within an ecotype (*t*-test, p≤0.001). Normalized fluorescence concentration is indicated on the y-axis. Data from three biological replicates for each ecotype were pooled and mean normalized for comparison.

hairs and to move these into the adjacent non-hair atrichoblast cell which enables the hair cell to acquire further solutes. The atrichoblast is in turn topologically poised to facilitate the optimized longitudinal movement of these nutrients along the root and up through the hypocotyl. This provides a functional division of labour in the two epidermal cell types, and may bridge the structure-function relationship in epidermal cell patterning owing to the global topological properties of this system.

Collectively, this experiment connects the topological analysis of global cellular connectivity using BC to molecular transport processes across the *Arabidopsis* hypocotyl.

## Comparison of topological features of diverse plant species

Plasticity in differential path length between epidermal cell types observed across *Arabidopsis* ecotypes suggests this system is not an intrinsic property of all hypocotyls. To determine the extent to which this higher-order property of global cellular organization is present in different contexts, two additional plants were selected for comparison with the Rosid species *Arabidopsis* belonging to the *Brassicacea* family. These were foxglove (*Digitalis purpurea*), an Asterid from the *Plantaginaceae* family, and the common poppy (*Papaver rhoeas*), a basal eudicot from the *Papaveraceae* family (*Figure 4A* and *Video 3*). These three species were selected based on their representation of diverse lineages of dicot angiosperms, enabling the topological investigation of multicellular complexity across distinct taxa. They are also all present in Western Europe and have annual, or in the case of foxglove biennial, life cycles. This enables comparisons within similar geographic distributions and life histories.

Both poppy and foxglove showed a significantly higher degree distribution in trichoblast cells than atrichoblasts (*Figure 4D*), consistent with all *Arabidopsis* ecotypes examined (*Figure 2D*). This property of local cell connectivity is therefore conserved across most wild-type genetic backgrounds examined (*Figure 4—figure supplement 1*).

The path length over atrichoblast cells is different in all species examined (*Figures 2E* and *4E*, *Figure 4—figure supplement 1*). Poppy epidermal cells lie upon significantly longer paths than both other species, while path length is shortest in *Arabidopsis* in these same cell types (*Figure 4—figure supplement 1*).

The quantitative distribution of BC is structurally and qualitatively similar between both epidermal cell types of poppy, yet a significantly shorter path length is present in trichoblasts (*Figure 4E*). This observation contrasts the presence of shorter atrichoblast path lengths observed in the Col and L*er* ecotypes of *Arabidopsis* (*Figure 2E*). While a significant difference in path length is detected between epidermal cell types in poppy, their relative contrast is less pronounced than between epidermal cell types in *Arabidopsis*, in addition to their path lengths being significantly longer diminishing the optimization of potential transport through these conduits (*Figure 2E*, *Figure 4—figure supplement 1*).

No significant difference in path length was detected between the two epidermal cell types of foxglove. Topologically this species is therefore similar to that of the *Arabidopsis* Cvi ecotype which also does not have a BC differential in the epidermis (*Figure 2E*).

Epidermal edge betweenness was greatest in *Arabidopsis* relative to each poppy and foxglove (*Figure 5F–G*), reflecting the shorter node path length present in this Rosid species relative to the others.

Path length between epidermal cell types is therefore plastic both within *Arabidopsis* ecotypes, and across different species. A progressive decrease in epidermal path length is observed from poppy, to foxglove, then *Arabidopsis*, and the contrast in path length between epidermal cell types is greatest in the Col and L*er* ecotypes of this latter species. This emergent property of reduced atrichoblast path length may represent a recent evolutionary innovation with regards to the optimization of transport along the exterior of the plant hypocotyl.

## Topological analysis of cellular patterning mutants

The diversity in morphological complexity which is observed in biological systems results from the intersection between design principles which have been selected for, and constraints which are imposed upon possible configurations (*Avena-Koenigsberger et al., 2015*; *Thompson, 1942*). Geometric, topological, mechanical and functional properties all act to limit the possible arrangements observed (*McGhee, 2006*). Differences in the topological properties of epidermal patterning suggest the global organization of this cell type is subject to modulation by genetically-encoded factors.

In plants, different cellular configurations are also possible within the same species (*Di Laurenzio et al., 1996*). The loss of genes mediating cellular patterning can yield viable offspring with alternative morphologies, and typically, compromised fitness. Characterizing the topology of cells in mutants where the causal genetic agent is known enables the contribution of gene activity towards the creation of topological properties to be investigated. Here we examined the properties of

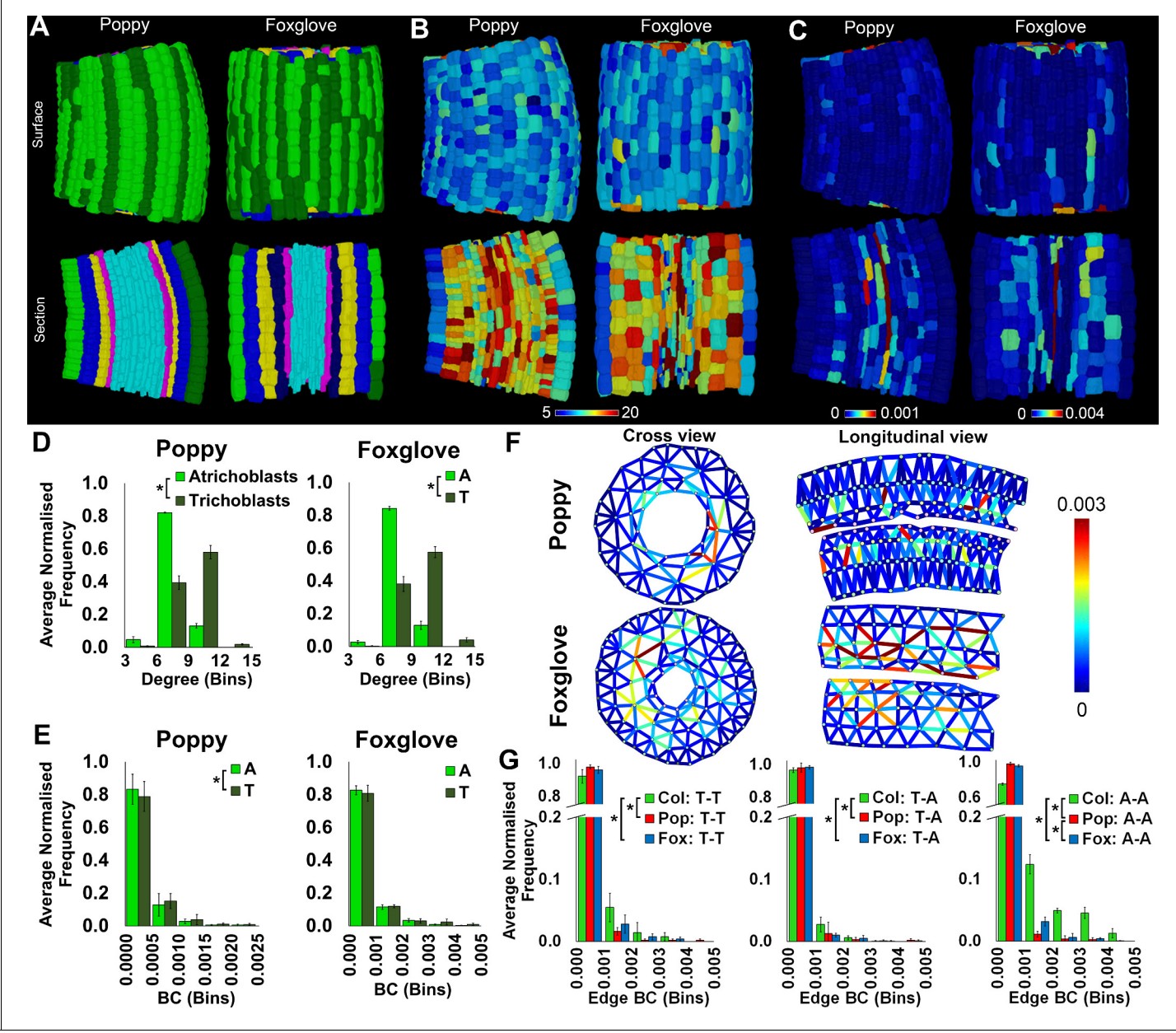

**Figure 4.** Cell type-specific topological characterization of hypocotyl cellular interaction networks from three plant species: Arabidopsis thaliana (Col), poppy and foxglove. (A) Surface and longitudinal section meshes of poppy and foxglove hypocotyls, with false color denoting cell type (dark green – trichoblast, light green – atrichoblast, blue – outer cortex, yellow – inner cortex second layer, navy blue – inner cortex third layer, pink – endodermis, cyan – vasculature). (B–C) Hypocotyl meshes with false color heat maps of (B) degree and (C) betweenness centrality (BC). (D) Average normalized frequency distributions of degree and (E) BC in epidermal cell types. (F) Edges describing hypocotyl cellular interactions false colored by edge BC. (G) Normalized frequency distribution of edge BC in the interfaces between different epidermal cell types (T – trichoblast, A – atrichoblast). Biological replicates were treated as individual samples and data represent the mean frequency of the bins across the triplicate samples. Error bars represent the standard deviation within a bin. An asterisk (*) represents significant difference between distributions using the chi-squared test for degree and the Kolmogorov–Smirnov test for BC, at the $p \leq 1.56 \times 10^{-5}$ level ($p \leq 0.05$ after Bonferroni correction for 3200 distribution comparisons).

The following figure supplement is available for figure 4:

**Figure supplement 1.** Cell type specific topological analysis of *Arabidopsis*, poppy and foxglove hypocotyl cellular arrangements.

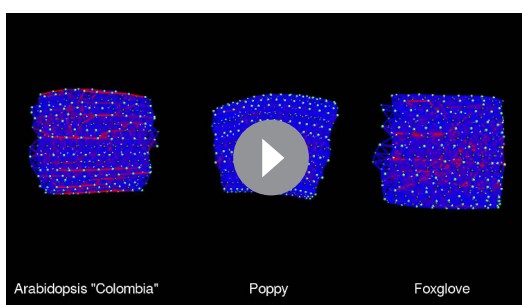

**Video 3.** Comparison of the hypocotyl cell connectivity networks of *Arabidopsis thaliana* Col (left), poppy (centre) and foxglove (right). Edge colour represents the edge random walk betweenness centrality (scale 0–0.0015, blue to red).

organ-wide hypocotyl cellular organization within three different cellular patterning mutants in *Arabidopsis*, selected based on previous reports of defects in the configuration of their cells.

*CYCLIN DEPENDENT KINASEA1;1 (CDKA1;1)* encodes a regulator of the cell cycle and mediates cellular patterning including asymmetric cell divisions (*Figure 5A* and *Video 4*) (*Nowack et al., 2012*). The *MONOPTEROS* (*MP*) gene encodes a transcription factor that regulates auxin-mediated gene expression and cellular patterning (*Hardtke and Berleth, 1998*). The *mp* mutant embryo lacks vasculature tissue at maturity and provides the opportunity to examine the impact these cell types play in controlling organ-level topology (*Figure 5A* and *Video 5*) (*Schlereth et al., 2010*). The *laterne* phenotype lacks embryonic leaves (cotyledons) in the mature embryo due to defects in localized auxin signaling (*Treml et al., 2005*) (*Figure 5A* and *Video 6*). The topological analysis of this mutant can link the activity of these mutations to patterning while exploring the role of non-cell autonomous signaling from the cotyledons in pattern formation within the hypocotyl during embryogenesis. The contribution of these genes towards the construction of the previously observed higher-order properties in epidermal patterning was examined by comparing their topological properties with their corresponding wild-type controls.

The *cdka1;1* mutant hypocotyl is qualitatively similar in appearance to the wild-type (*Figure 5A* and *Video 4*), and shows a similar spatial distribution and frequency of degree to its control accession Col (*Figure 5B,D* and *Figure 5—figure supplement 1*) (*Dissmeyer et al., 2009*). The number of local connections in each trichoblast and atrichoblast cells did not show any significant differences between this mutant and wild type, with trichoblast cells being more highly connected. The difference in path length between the two epidermal cell types is however lost in the *cdka1;1* mutant (*Figure 5D*), while a significant reduction in endodermal path length is also observed (*Figure 5—figure supplement 1*). This demonstrates that the patterning mediated by the *CDKA1;1* gene product is responsible for the generation of reduced path length within the atrichoblast cell files of the epidermis, and also limits the construction of short paths in the endodermis. This further links the activity of this gene to both local and global patterning consequences within a specific cell type in this alternate cellular configuration.

The trichoblast degree distribution of mature *mp* embryos is similar to its Col wild-type equivalent while atrichoblasts have more local connections (*Figure 5B,E*). Node BC is however much higher in both epidermal cell types of the mutant, and these cells lie upon significantly shorter paths (*Figure 5E*). This demonstrates that *MP*-mediated patterning acts to generate longer paths in the hypocotyl epidermis during embryogenesis, but is not required for the path length asymmetry between the two constituent cell types. The consequences of lacking a vascular system in this mutant can be observed through the visualization of the spatial distribution of edge betweenness (*Figure 5G*). Here a radial route of short path length interfaces between cells is observed through the center of this cellular assembly, a property qualitatively absent in other hypocotyl cellular interfaces.

The topological analysis of the *laterne* mutant revealed no significant difference in degree across all cell types, with the exception of the endodermis (*Figure 5F*, *Figure 5—figure supplement 1*). Each epidermal cell type lies upon longer paths than in its L*er* wild-type equivalent, yet the relationship in path length between the epidermal cell types is the same as the control with atrichoblasts lying upon shorter paths than trichoblasts (*Figure 5F*, *Figure 5—figure supplement 1*). The *laterne* phenotype therefore limits global path length in the hypocotyl epidermis and may be controlled by non-cell autonomous signaling from the cotyledons, or be a local consequence of the cellular patterning activity of these gene products during embryogenesis (*Treml et al., 2005*).

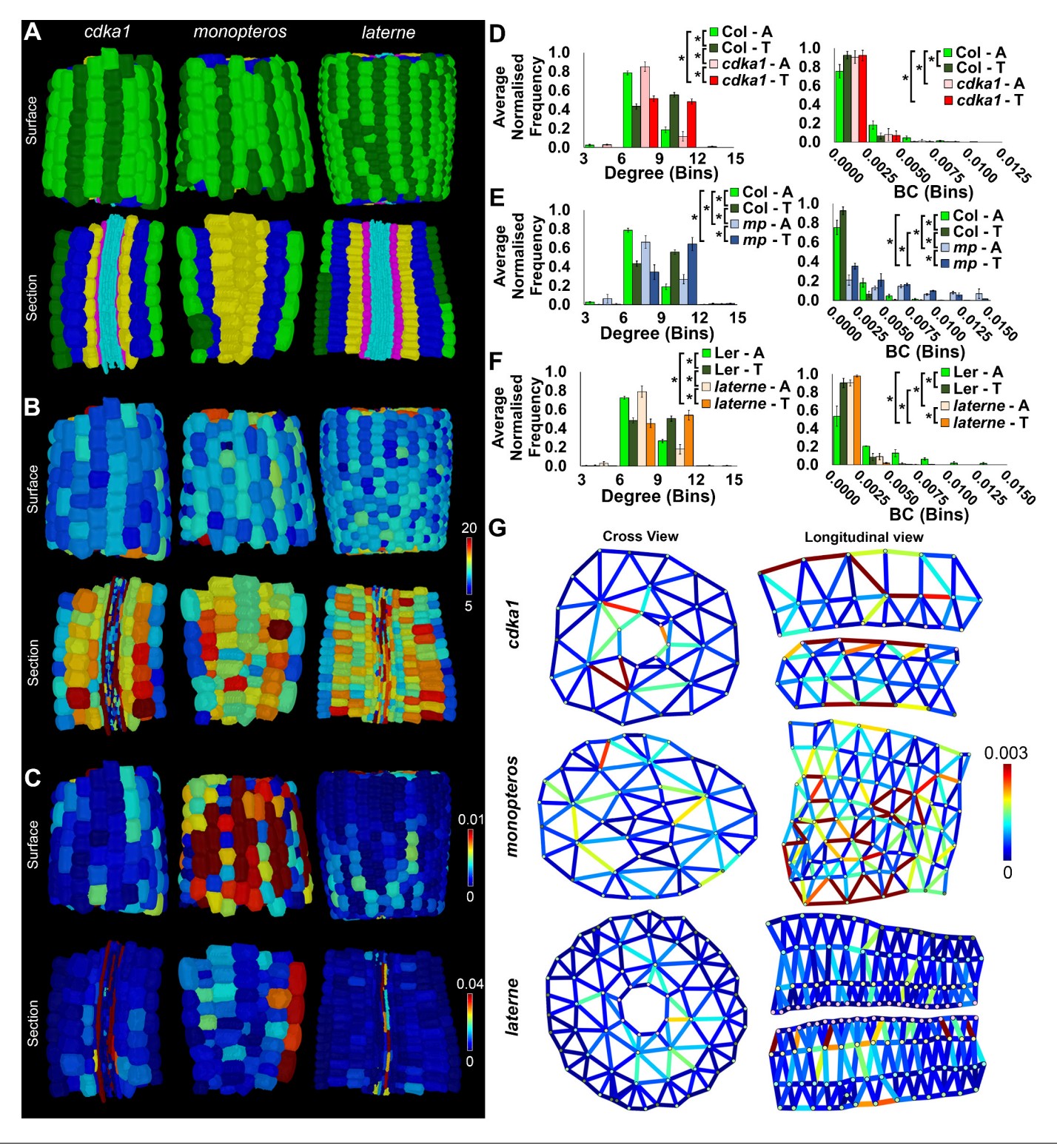

**Figure 5.** Comparisons of cell type-specific patterning and path length in the hypocotyls of wild-type *A. thaliana* (Col) and three *A. thaliana* mutants: *cdka1;1*, *monopteros* and *laterne*. (A) Surface and longitudinal section meshes of the three mutants, with false color cell type annotation as in *Figure 2*. (B–C) Mutant hypocotyl meshes with false color heat maps of (B) degree and (C) betweenness centrality (BC). (D) Degree and BC in the *cdka1;1* mutant and corresponding Col wild-type control. (E) Degree and BC in the *mp* mutant and corresponding Col wild-type control. (F) Degree and BC in the *laterne* mutant and corresponding L*er* wild-type control. (G) Edges describing cellular interactions false colored by edge BC in mutant hypocotyls. Biological replicates were treated as individual samples and data represent the mean frequency of the bins across the triplicate samples. Error bars

*Figure 5 continued on next page*

*Figure 5 continued*

represent the standard deviation within a bin. An asterisk (*) represents significant difference between distributions using the chi-squared test for degree and the Kolmogorov–Smirnov test for BC, at the $p \leq 1.56 \times 10^{-5}$ level ($p \leq 0.05$ after Bonferroni correction for 3200 distribution comparisons).

The following figure supplement is available for figure 5:

**Figure supplement 1.** Cell type specific topological analysis of mutant *Arabidopsis* hypocotyl cellular arrangements.

The topological analysis of global cellular patterning in genetic mutant backgrounds allows the quantitative contribution of gene activity to be the linked to the formation of complex cellular assemblies, and their global organizational properties at cell type specific resolution. These analyses also enable the exploration of extant topological morphospace within plant organs at cellular resolution (*Avena-Koenigsberger et al., 2015*; *Ollé-Vila et al., 2016*). These alternative cellular topologies represent additional cellular configurations which are both geometrically and topologically possible, yet most likely compromised in their fitness, and thus function. Understanding the properties of these alternative configurations and their associated functional fitness is key to uncovering the structure-function relationship at the level of global cellular organization, and how these properties are genetically encoded.

## The relationship between geometric, local and global topological features of cells and cell networks

To understand how cell geometry is related to topology, mutual information measured in Shannon entropy was calculated within all samples used in this study. This determines the extent to which two variables are related to one another (*Kraskov et al., 2004*). A high Shannon entropy between two entities demonstrates a high level of relatedness, and the ability of one variable to predict the value of the other.

A similar pattern of mutual information was observed across all cell types and samples (*Figure 6*, *Figure 6—figure supplements 1–2*). Each cell area and volume share a high degree of similarity, while mutual information between each area and degree, and degree and BC, show much less in common (*Figure 6*, and *Figure 6—figure supplements 1–2*). Each area and BC, and volume and BC, have minimal mutual information, demonstrating that the global property of path length is not related to cell size. Likewise, the number of neighbours a cell has is weakly associated with the length of the path it lies upon. The control of path length in the hypocotyl is therefore largely independent of each cell size and the number of neighbours it has, and represents a property emerging from the global context within which a cell is situated in an organ. This conclusion is supported by the lack of significant differences in degree distributions which in turn show significant differences in path length distribution (*Figure 2—figure supplement 1*, *Figure 4—figure supplement 1*).

## Variability in cellular patterning

The spatial distribution of degree and BC is unevenly distributed within individual cells of a given cell type in all samples examined (*Figures 2* and *4–5*). This highlights the presence of topological variability in cell organization within these organs. We explored the extent of the variability present in BC in individuals across the *Arabidopsis* ecotypes, different species and cellular patterning mutants.

The coefficient of variation for BC for each cell type was plotted to examine the extent of the variability in the creation of path lengths between individual hypocotyls (*Figure 7A–B* and *Figure 7—figure supplement 1*). Both epidermal cell types across all samples examined

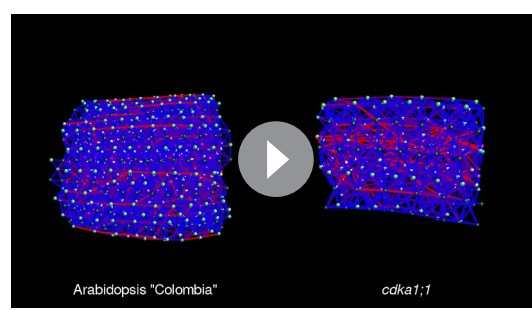

**Video 4.** Comparison of the hypocotyl cell connectivity networks of wild-type *Arabidopsis thaliana* Col (left), and the mutant *cdka1;1*. Edge colour represents the edge random walk betweenness centrality (scale 0–0.0015, blue to red).

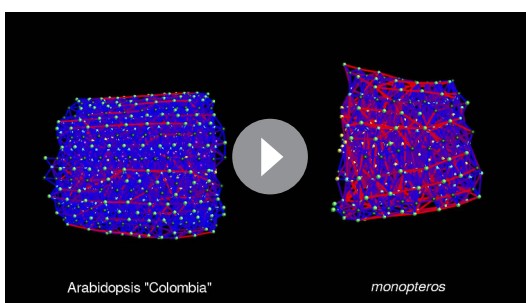

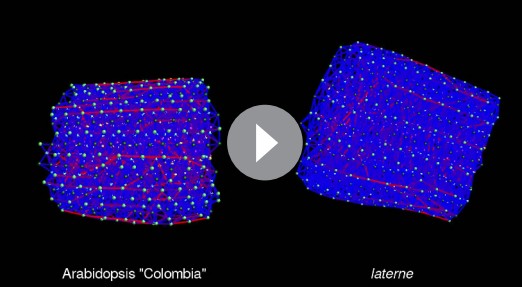

**Video 5.** Comparison of the hypocotyl cell connectivity networks of wild-type *Arabidopsis thaliana* Col (left), and the mutant *monopteros*. Edge colour represents the edge random walk betweenness centrality (scale 0–0.0015, blue to red).

**Video 6.** Comparison of the hypocotyl cell connectivity networks of wild-type *Arabidopsis thaliana* Col (left), and the mutant *laterne*. Edge colour represents the edge random walk betweenness centrality (scale 0–0.0015, blue to red).

showed no significant differences in their robustness to generating consistent path lengths relative to one another, with the exception of the *mp* mutant which had a greater extent of variability (*Figure 7A–B*). Similar results were observed in the cortical cell layers of this mutant relative to other genotypes examined (*Figure 7—figure supplement 1*). The *MONOPTEROS* gene product therefore promotes patterning robustness and the generation of consistent path lengths in *Arabidopsis* hypocotyls with regards to organ-level cell topology.

## Global view of multicellular complexity across cell types in plant hypocotyls

Beyond the epidermis, cell type-specific data describing local connectivity and path length in other cell types were generated. Their analysis provides the opportunity to understand the overarching principles underlying the assembly of cells in this organ across diverse genetic backgrounds and species.

Across *Arabidopsis* genotypes, degree distributions show no significant differences in either cortical cell layers or the endodermis (*Figure 2—figure supplement 1*), indicating a robust control in local connectivity within the internal cell layers of this species. Fewer significant differences in BC distribution are seen from the outer to inner cell layers, with no significant differences being observed in the endodermis across ecotypes (*Figure 2—figure supplement 1*). Degree distribution also shows no significant difference across all cell types in foxglove, poppy and *Arabidopsis* Col hypocotyls, with the exception of the endodermis (*Figure 4—figure supplement 1*).

BC distribution is the same in all species within in the inner cortex, however the endodermis shows some significant differences highlighting the presence of plasticity in path length generation (*Figure 4—figure supplement 1*). In contrast, the outer cortex showed similar path lengths across all species investigated.

To better understand the relationship between the topological properties of different cell types and how both degree and BC change across the radial arrangement of the hypocotyl, scatterplots of each mean and median average value, respectively, were generated to visualize these relationships (*Figure 8A–E*). Path length remains relatively constant within both epidermal cell types across species and genetic backgrounds, however degree shows a relatively broader range (*Figure 8A–B*). The degree to BC relationship in the inner cortex contrasts this with mean degree being relatively constant across genetic backgrounds, and path length showing a greater relative spread in values (*Figure 8D*). Each the outer cortex and endodermis have a relatively broad range of degree and BC across the samples analysed. These observations suggest that the epidermal cell types are topologically distinct, and that the outer cortex and endodermis are more similar to one another than to the inner cortex. In all instances *monopteros* samples showed a broad distribution of degree and BC values, highlighting the topologically unique nature of this mutant and a lack of robustness in its global pattern formation (*Figure 7*).

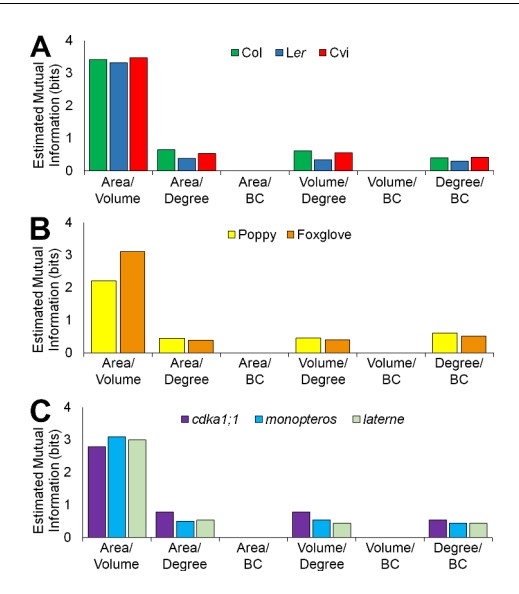

**Figure 6.** Mutual information between geometric and topological cell features in the hypocotyl epidermis. (A–C) Estimated mutual information between area, volume, degree and betweenness centrality of epidermal cells in (A) three *A. thaliana* ecotypes, Col, L*er* and Cvi, (B) poppy and foxglove, and (C) three *A. thaliana* mutants, *cdka1;1*, *monopteros* and *laterne*. Mutual information was measured using bits of Shannon entropy and is plotted along the y-axis.

The following figure supplements are available for figure 6:

**Figure supplement 1.** Mutual information between geometric and topological hypocotyl cell features in all cell types, atrichoblasts, and trichoblasts.

**Figure supplement 2.** Mutual information between geometric and topological hypocotyl cell features in the outer cortex, inner cortex and endodermis.

**Figure supplement 3.** Mutual information between edge features across the hypocotyl cell connectivity network.

While informative, the use of mean values as a summary statistic in scatterplots fails to capture the breadth of the data represented within a distribution. To address this we performed pairwise statistical tests for each degree and BC distribution, respectively, for all cell types to determine whether their topological properties were significantly similar or not. The test statistic obtained by comparing two distributions (chi-squared value for the discrete variable degree, Kolmogorov–Smirnov test value for the continuous variable betweenness centrality) was used as a measure of the distance between the distributions. These distances were used to generate a distance matrix, enabling their clustering.

Significant relationships between degree distributions were calculated and were placed into three clusters (*Figure 8F*). Each wild-type atrichoblast and trichoblast cells clustered independently (*Figure 8G*). All other cell types were present in the third group, with the exception of the poppy endodermis. This revealed that each epidermal cell type has a distinct local connectivity, while all other internal cell types share similar properties in this regard. Degree alone may therefore distinguish different epidermal cell types, and cells internal to the epidermis.

The clustering of significant similarities between BC distributions enabled the relationships between the higher-order properties of diverse cell types across genetic backgrounds to be established. Three clusters were selected to categorize these pairwise similarities (*Figure 8H*). All wild-type trichoblasts, with exception of poppy, cluster together (*Figure 8I*). This is likely a consequence of the stringent regulation of path length we previously identified to be present within this cell type, and increased path length within the poppy epidermis (*Figure 2—figure supplement 1*, *Figure 4—figure supplement 1*). Conversely, atrichoblasts have greater plasticity in their path length across genetic backgrounds (*Figure 2—figure supplement 1*, *Figure 4—figure supplement 1*) and are more broadly distributed across the clustering diagram. *Arabidopsis* Cvi and Foxglove atrichoblasts remained in the trichoblast cluster as these do not show a significant path length differential between them (*Figures 2E* and *4E*). These relationships between epidermal cell path lengths reflect the previously observed higher-order properties of this cell type.

All wild-type outer cortical cells clustered together along with *Arabidopsis* endodermal cells, which have similar path lengths (*Figure 8I*), supporting earlier conclusions relating to the topological relationship between these two cell types. Inner cortical cells also clustered together highlighting their topological uniqueness within the hypocotyl.

Collectively these results suggest that each epidermal cell type is topologically distinct from all other cell types, as is the inner cortex. The outer cortex and endodermis share global organizational similarities. These observations reveal general principles of multicellular architectural design within the wild-type plant hypocotyl. It is through these higher-order path length principles that the

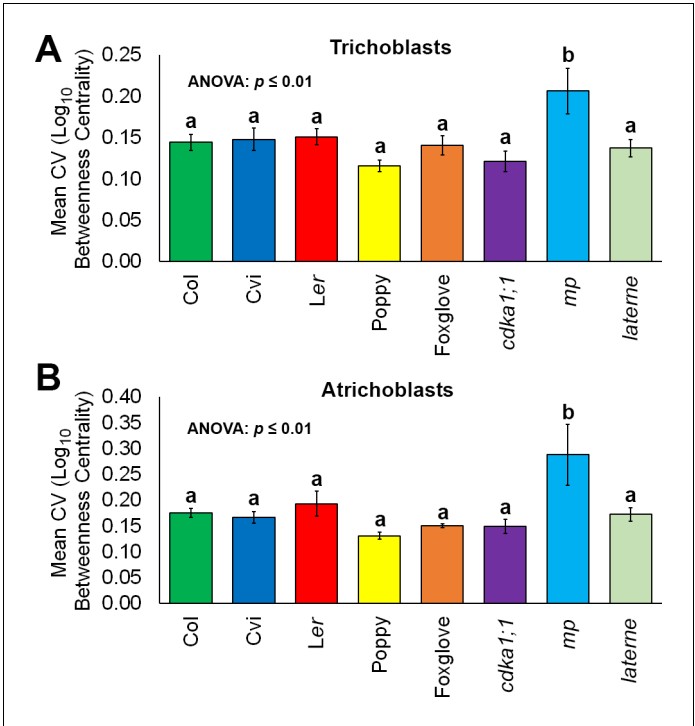

**Figure 7.** Robustness in cellular patterning and topological properties in the plant hypocotyl. Comparisons of the coefficient of variation (CV) of betweenness centrality in different cell types of hypocotyls from *A. thaliana* ecotypes (Col, L*er* and Cvi), mutants (*cdka1;1*, *monopteros* and *laterne*) and other plant species (poppy and foxglove). (**A**) Mean CV in trichoblast cells. (**B**) Mean CV in atrichoblast cells. Data represented are the means and standard deviation of three biological replicates. ANOVA tests suggested significant differences between groups ($p \leq 0.01$) and an asterisk (*) indicates significant difference to all other groups (Tukey's test, $p \leq 0.01$).

The following figure supplement is available for figure 7:

**Figure supplement 1.** Variability in hypocotyl betweenness centrality.

movement of molecules occurs, bridging a structure-function relationship between cellular patterning and the molecular processes which unfold within complex plant organs. Understanding these principles and the ability to calculate them at single cell resolution enables the mathematical prediction of intercellular and optimized molecular flux and intercellular communication which underlies plant growth and development.

## Discussion

Extensive research into cellular patterning has been undertaken previously in diverse biological systems. In both plants and animals, rules governing the placement of cell division planes in a default state have been described (*Besson and Dumais, 2011*; *Hofmeister, 1867*) and are sufficient to explain pattern formation in contexts including the shoot apical meristem of plants (*Sahlin and Jönsson, 2010*; *Shapiro et al., 2015*) and *Drosophila* development (*Gibson et al., 2011*). Deviations from this default rule have been considered to be actively regulated cell divisions that lead to the generation of novel patterns. In the context of plant embryo development, this has been reported to be mediated by the hormone auxin (*Yoshida et al., 2014*) and downstream factors controlling auxin-mediated patterning have been described to be involved in diverse aspects of this formation (*Schlereth et al., 2010*). In addition to genetically controlled events, a role for mechanical feedbacks in the control of cellular patterning have also been reported in plants (*Hamant et al., 2008*; *Louveaux et al., 2016*).

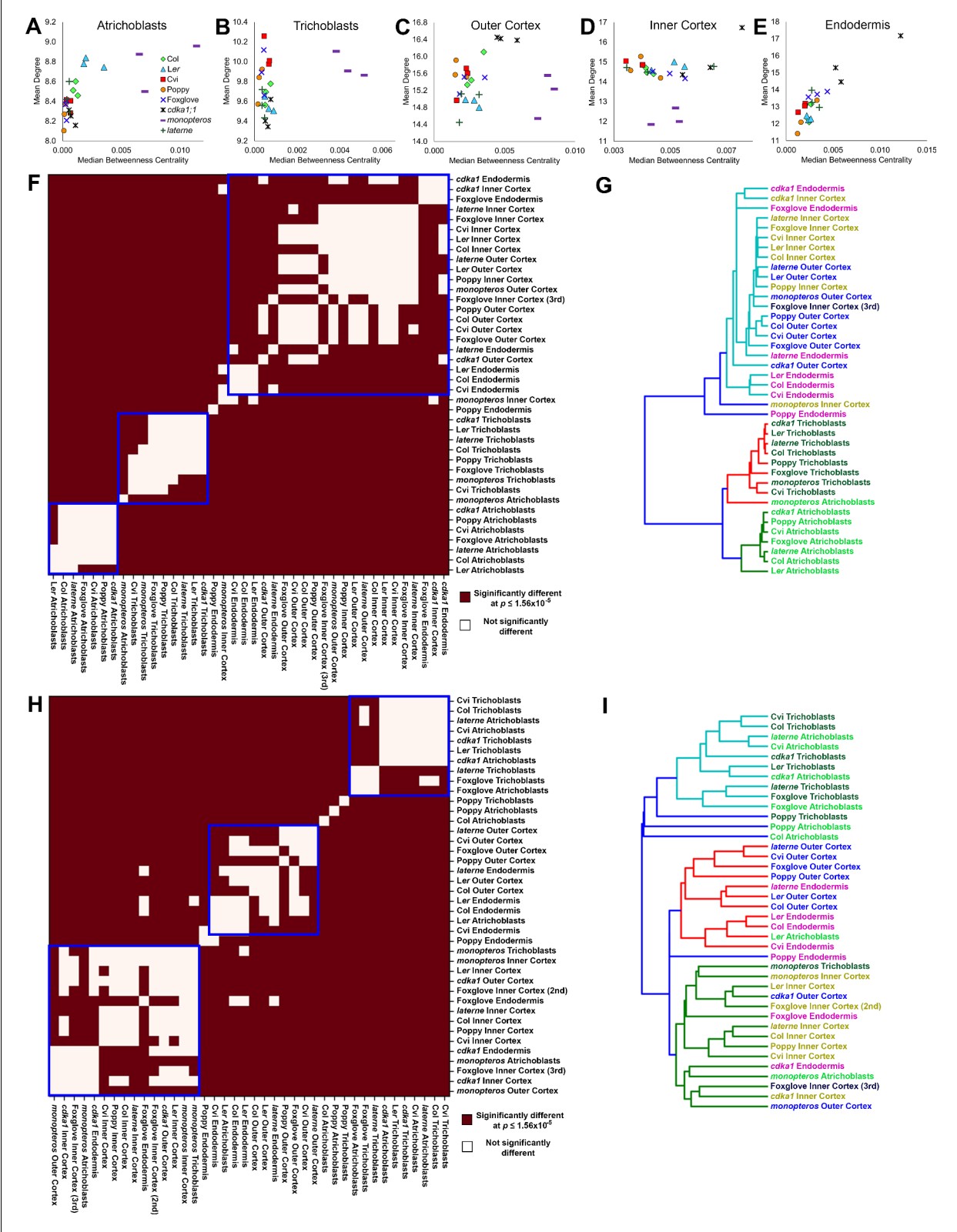

**Figure 8.** Cell type-specific topological principles of hypocotyl multicellular architecture. (A–E) Scatterplots of mean degree and median betweenness centrality in (A) trichoblasts, (B) atrichoblasts, (C), the outer cortex, (D) the inner cortex and (E) the endodermis for each sample used in this study. (F–G) Analysis of degree distribution relationships using the inverse of the chi-squared test statistic as a measure of distance. (F) pairwise significance tests for

*Figure 8 continued on next page*

*Figure 8 continued*

degree distribution and (**G**) a dendrogram of the data from (**F**). (**H–I**) Analysis of betweenness centrality distributions using the inverse of the Kolmogorov–Smirnov test statistic as a measure of distance, with (**H**) pairwise significance tests and (**I**) a dendrogram of the data from (**H**).

While the mechanisms underlying the creation of cellular patterns have been studied widely, bridging their structure-function relationship, and understanding of the functional role of these patterns in organ biology remains less clear. The network-based analysis of cellular patterning can bridge this gap by enabling cellular patterning to be captured and quantified. This approach was originally applied in the field of neuroscience where global neuronal connectivity was mapped in the worm *C. elegans* (*White et al., 1986*). This approach has since been employed to more complex nervous systems in the growing field of connectomics (*Bullmore and Sporns, 2009*). Methodological imaging and computational advances are now leading to the generation of cellular resolution connectomes in complex brains (*Economo et al., 2016*), though that of *C. elegans* remains the only complete connectome to date.

The immobility of cells within plant organs simplifies the topological analysis of multicellular arrangements as cellular associations are maintained throughout development. The application of this approach to other biological systems where cell migration occurs is also possible, and would involve the analysis of 3D cellular interaction dynamics (*Heller et al., 2016*). The transient nature of edges in these systems will dynamically alter the topology of these cellular interaction networks. Analytical frameworks to analyze temporal network datasets such as these have been developed (*Holme and Saramäki, 2012*).

The analysis of animal epithelium development has also been studied using a network-based topological approach. Local epithelium cellular connectivity was sufficient to distinguish different species and organs (*Escudero et al., 2011*). A separate study in *Drosophila* imaginal discs revealed cell degree to be tightly regulated during the development of this tissue (*Gibson et al., 2006*). These same authors then extended this approach to identify a bias in the placement of the cleavage plane following the degree of adjacent cells during pattern formation, a phenomenon also present in the shoot apex of plants (*Gibson et al., 2011*). In each of these studies, local connectivity was examined, with the global higher-order principles of organ design remaining unexplored.

By viewing organs as a complex system of interacting cells, this study applied a systems-based approach to understand the quantitative principles of multicellularity and uncover their higher order properties. The capture and analysis of organism-wide cellular connectivity enabled both the local and global topological features of complex organism design to be investigated, revealing how multi-cellular assemblies operate as an integrated system and their emergent properties.

The application of cell-type specific topological analysis using BC identified the presence of a previously undescribed optimized transportation route in the atrichoblast epidermal cell files of the *Arabidopsis* hypocotyl in the Col and L*er* ecotypes (*Figure 2E*). This cell type is uniquely poised to regulate information flow along the hypocotyl by lying on shorter paths, despite having fewer local neighbours (*Figure 2D–E*). This property was not present in *Arabidopsis* Cvi, nor in other species studied including poppy and foxglove (*Figure 4E*). This previously undescribed higher order property of epidermal patterning is therefore plastic across diverse genetic backgrounds.

The functional significance of these reduced path lengths was demonstrated by tracking the movement of the fluorescent small molecule fluorescein through *Arabidopsis* seedlings (*Figure 3*). BC predicted the bulk flow of these small molecules through diverse epidermis topologies at single cell resolution. These patterns therefore provide optimized routes for molecular trafficking through organs, though they are not obligatory for organ function in light of the plastic nature of their presence. The reduction of epidermal path length may support rapid plant growth, or represent a recent evolutionary innovation of higher-order multicellular optimization that has not yet been widely adopted across diverse genetic backgrounds.

The topological analysis of patterning mutant hypocotyls revealed the quantitative contribution of genetically-encoded factors towards the creation of higher-order features at cell type-specific resolution (*Figure 5*). The construction of reduced atrichoblast path length was established to be mediated by the patterning activity of the *CDKA1;1* gene, and overall epidermal path length is limited by *MONOPTEROS* and promoted in *laterne* (*Figure 5D–F*). This approach provides a step change

beyond qualitative descriptions of altered local cellular organization, and enables quantitative links between the molecular and cellular scales of complexity to be bridged.

The genetic contribution towards the regulation of consistent global path length in the hypocotyl epidermis also revealed an additional link between the activity of patterning genes and control over robustness in the generation of higher order path length (*Figure 7*). This provided novel insight into the role of *MONOPTEROS* in the control of topological robustness in cellular patterning during embryogenesis. These observations further demonstrated that while epidermal path length is plastic between genetic backgrounds, it is not variable across individuals.

The structural networks generated in this study may be considered the roadmaps upon which molecular events occur within the hypocotyl. The annotation of these structural templates with additional molecular information will lead to the creation of functional networks, which will provide multi-dimensional insight into the molecular complexity which unfolds in the context of multicellular organ development.

Uncovering the organizing principles (*Sena and Birnbaum, 2010*) of multicellular assemblies in extant organs provides insight into the cellular configurations which are each selected for in wild-type plants, and which are possible in the case of patterning mutants (*Avena-Koenigsberger et al., 2015*). These cellular interaction networks result from pattern formation, and represent the topological output of the self-organizing processes which underlie plant organ formation.

Uncovering nature's multicellular structural design principles and the functions they encode will provide fundamental insight into the evolutionary forces driving increased complexity and optimization following the transition to multicellularity (*Ollé-Vila et al., 2016*). This enhanced understanding of these properties establishes a framework for the rational re-design of multicellular configurations and organ functionality.

## Materials and methods

### Plant growth conditions

*Arabidopsis* wild-type plants of ecotypes Colombia, Cape Verdi Islands (Cvi) and Landsberg *erecta* (L*er*) and associated mutants *laterne* (*Treml et al., 2005*), *monopteros* (*mp* B4149) (*Schlereth et al., 2010*) and *cdka1;1* (ProCDKA;1::CDKA;1-T14D;Y15E) (*Dissmeyer et al., 2009*) were grown in environmentally controlled cabinets using 16 hr light (light intensity 150–175 μmol m$^2$s$^{-1}$ at 23°C and 8 hr dark at 18°C). Plants with dry siliques were harvested and seeds cleaned through a 500 μm mesh. Poppy (*Papaver rhoeas*) seeds were grown under field conditions, and foxglove (*Digitalis purpurea*) was collected from the Birmingham Botanical Gardens.

### Sample preparation and image acquisition

Embryos were dissected from germinating seeds with a scalpel and forceps using a binocular microscope at 3 hr following their imbibition. Samples were prepared for confocal microscopy as described previously (*Bassel et al., 2014*; *Truernit et al., 2008*) and imaged using a Zeiss LSM 710 confocal microscope.

### Cellular annotation and cellular interaction network extraction

This was performed as described previously (*Montenegro-Johnson et al., 2015*). Cell type designation was achieved by using CellAtlas3D, followed by manual corrections. Epidermal cell types were assigned based on their positions above underlying cortical cell layers, with trichoblasts bridging multiple underlying cortical cell files. After using 3DCellAtlas, further manual annotation was based on position in the hypocotyl and cell shape. Shape became particularly important in assigning cell types in *cdka1;1* and *monopteros* mutant samples, where endodermal cells are not present throughout the entire hypocotyl.

### Network analysis

Cellular interaction networks were exported from MorphoGraphX as text files with shared associations between cells as edge weights in μm$^2$. Edge lists were imported into the Python package NetworkX (*Hagberg et al., 2008*) where analyses were performed. Text files were in turn imported into MorphoGraphX as heat maps to visualize node properties in situ by false coloring segmented cells.

Edge effects in topological analysis due to imaging boundaries were minimized by including all cells in topological analyses, but only examining the properties of those in the central regions (*Figure 1—figure supplement 1*). Vascular cells were captured and included in topological analyses, though quantitative data were not reported due to the presence of sporadic cellular segmentation errors. Biological triplicates were used across this study.

To account for imaging artifacts affecting cellular connectivity, edges representing the connection of two adjacent cells which shared less than 2 µm² cell wall area were removed between all cell types except the vasculature (*Figure 1—figure supplement 1*). BC for mean and standard deviation calculations was $\log_{10}$ transformed to generate normal distributions. 3D networks were visualized using BioLayout3D (*Theocharidis et al., 2009*). All network data are available in Source Dataset.

### Network normalization

To account for differences in network size across samples, node betweenness centrality was normalized by $2/((N-1)(N-2))$, and edge betweenness centrality was normalized by $2/N(N-1))$ where $N$ is the number of nodes in a network. Due to inherent constraints of physical Euclidean space, centrality was not normalized by network density in the hypocotyl samples.

### Network weighting

Where networks were weighted, the inverse of the absolute connecting cell wall area (µm²) was used in the calculation of weighted betweenness centrality (*Newman, 2010*).

### Statistical tests

Estimated mutual information (Shannon entropy) was calculated using the Non-Parametric Entropy Estimation Toolbox (NPEET), a Python package that incorporates functions for both continuous (betweenness centrality, area, volume) and discrete (degree) variables (*Kraskov et al., 2004*).

Two-sample *t*-tests, chi-squared tests and two-sample Kolmogorov-Smirnov tests were performed using Python and the SciPy library (*Jones et al., 2001*). ANOVA with Tukey's HSD post-hoc tests were carried out using SPSS. Hierarchical clustering was performed using Python and the scikit-learn package (*Pedregosa et al., 2011*). The Bonferroni correction for multiple comparisons was used to correct the *p*-value in every chi-sqaured test and Kolmogorov-Smirnov test, based on the 3200 pairwise comparisons used throughout the study.

### Fluorescein transport assay

The three shown ecotypes were germinated in 1/2 MS plates in complete darkness for 4 days, then exposed for 2.5 hr to 0.8% agar plates containing 50 µM fluorescein (Sigma Aldrich). Before imaging by fluorescence confocal microscopy, samples were treated with (10 mg/mL) propidium iodide to visualize cell walls. 3D cell segmentation and quantification of fluorescence density within each cell was performed using FIJI (*Schindelin et al., 2012*) and MorphoGraphX (*de Reuille et al., 2015*). Cells were meshed using a cube size of 2 and no smooth passes were used. Fluorescein concentration within individual cells was calculated using the Heat Map function with Volume and Internal Signal selected. Data presented consist of at least 40 segmented cells from three independent biological replicates for each ecotype.

## Acknowledgements

We thank Tom Freeman for support with BioLayout3D, Nico Dissmeyer, Dolf Weijers and Ramon Angel Torres Ruiz for mutant seeds. We also Richard Smith and Soeren Strauss for support with MorphoGraphX, and Iain Johnston for helpful discussions on statistics. GWB and PS were supported by BBSRC grant BB/J017604/1, GWB by BBSRC Grant BB/L010232/1, GWB and SD-N by Leverhulme Trust Grant RPG-2016–049, GWB and HX by BBSRC grant BB/N009754/1 and MDBJ was supported by BBSRC DTP BB/M01116X/1 MIBTP.

# Additional information

## Funding

| Funder | Grant reference number | Author |
|---|---|---|
| Biotechnology and Biological Sciences Research Council | BB/J017604/1 | Petra Stamm<br>George W Bassel |
| Leverhulme Trust | RPG-2016-049 | George W Bassel |
| Biotechnology and Biological Sciences Research Council | BB/L010232/1 | George W Bassel |
| Biotechnology and Biological Sciences Research Council | BB/N009754/1 | Hao Xu<br>George W Bassel |
| Biotechnology and Biological Sciences Research Council | BB/M01116X/1 | Matthew DB Jackson |

The funders had no role in study design, data collection and interpretation, or the decision to submit the work for publication.

## Author contributions

MDBJ, Resources, Data curation, Software, Formal analysis, Investigation, Visualization, Writing—review and editing; HX, Resources, Data curation, Investigation; SD-N, Resources, Data curation, Formal analysis, Investigation; PS, Resources; GWB, Conceptualization, Data curation, Formal analysis, Supervision, Funding acquisition, Investigation, Methodology, Writing—original draft, Project administration, Writing—review and editing

## Author ORCIDs

Matthew DB Jackson, http://orcid.org/0000-0003-0225-8235
Salva Duran-Nebreda, http://orcid.org/0000-0002-2539-3539
George W Bassel, http://orcid.org/0000-0002-3434-4499

# Additional files

## Supplementary files

• Supplementary file 1. Source dataset: Text files of network data (edge lists) describing cell connectivity and cell type annotations used in this study. Cells are represented by arbitrary labels, and interaction size ($\mu m^2$) and cell types for each cell-cell interaction are included in additional columns. Details are explained in the associated readme file. Biological triplicate samples for each genetic background are labelled sequentially.

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
