## [Decision Letter]

Thank you for submitting your article "Topological analysis of multicellular complexity in the plant hypocotyl" for consideration by *eLife*. Your article has been favorably evaluated by Christian Hardtke (Senior Editor) and three reviewers, one of whom is a member of our Board of Reviewing Editors. The reviewers have opted to remain anonymous.

The reviewers have discussed the reviews with one another and the Reviewing Editor has drafted this decision to help you prepare a revised submission.

Summary:

In this work Jackson et al. provide a general framework to study cellular networks in plant tissues. They show that different cell types have different network properties in *Arabidopsis*, and that this holds, to some extent, across different *Arabidopsis* ecotypes and (to a lesser extent) between species. These results present an interesting methodological advance, and the potential to uncover previously unseen patterns, if a number or technical and presentation issues are addressed.

Major issues:

1) *eLife* papers describing new techniques or analysis approaches like this one should be accessible and interesting to a wide audience. Several reviewers felt that the authors did not yet make a persuasive case that they are discovering general principles here, and for this work to be useful for the developmental biology community, the connection between the detailed measurements and biological problems must be made very clearly. This "Major issues" section enumerates what they considered to be necessary changes to be made in the revision:

A) The connection between what is measurable and what can be interpreted as the biological significance of the measurement needs to be done with some care. For example, the authors interpret BC as meaning that "atrichoblast cell files are topologically poised to mediate the optimized movement of information" and "having the capacity to control information movement through a limited number of local interacting partners". This is a leap. For example, BC may highlight nodes that form the sole pathway between two otherwise disconnected parts of the network. In that case the argument that BC signifies the potential to control flow (of information, or anything else) is justified. However, BC can be high in many other contexts. In the cellular contact network in particular there will be many ways that information can pass between cells, as this network resembles a lattice. A high BC node therefore could lie on a slightly shorter path, but there will be many other paths, perhaps even of similar length. The equivalence between high BC and a capacity to control is therefore much less obvious. The authors should put less emphasis on this interpretation of BC. They may still have a point here, but at present the manuscript is overstating the connection between BC and control. This is linked to a general feeling that the equivalence of cellular networks and networks of directed information flow (based on the evidence presented in this paper), is overstated.

B) Several reviewers felt there was not a clear path between the technique described here and genuine insight into developmental biology. The clearest way to show that this technique can lead to new biological insights is to provide such an insight. This may be achieved by revisiting old experiments in the literature that had some contradictory or hard to understand elements and showing how the approach presented in this paper can explain the discrepancy/discordance in the data. Alternatively, an observation made here should be predictive of a behavior; for example, the authors describe a difference in connectivity between atrichoblasts or trichoblasts: what would you predict are different properties of a hormone or Ca or physical signal sent through one or the other of these cell types? And how would you confirm this experimentally?

C) The rationale explaining how the authors picked the different plant species and the different mutants are not crystal clear. For the flow of the ideas and the understanding of the reader, it would be nice to include two paragraphs explaining the general concept and why the authors chose these plant species (first paragraph) and mutants (second paragraph).

D) There needs to be a more coherent section at the end summarizing precisely what principles hold across ecotypes, species, and mutants (backed up by statistical significance measures). A more rigorous set of conclusions of this kind would provide an adequate justification for the rather strong and general claims made in the abstract. The question that needs to be clearly answered in this summary is: What can network measures predict about plant cells?

E) There are some technical issues in the way data are presented or interpreted. For example, weighted BC (subsection “Topological analysis of the wild-type Colombia *Arabidopsis* hypocotyl”, tenth paragraph). Generalisations of BC to weighted networks are not entirely straightforward, as there is more than one way to interpret path length in weighted networks. One way for example is to use the reciprocal of the weight as the contribution that a link makes to the overall path length. The authors should clarify what version of weighted BC they are using. It seems that the inverse of the connecting cell wall area was used as the weight itself, but weighted path length calculations often use the reciprocal of the weight, which means that there might be a double reciprocal here? In any case the definition of the weight and the exact algorithm should be made explicit.

2) In general, arguments throughout the manuscript rely a lot on visual comparisons of distributions, which are not always convincing. There are plenty of rigorous statistical tools for comparing two distributions, which would return a measure of significance for the difference between the distributions. Calculating such significance values for all the relevant comparisons of distributions in the paper would make the authors' arguments much stronger (providing the statistics support their claims). In addition, since you are able to study many different parameters extracted from the plant tissues (e.g. cell characteristics, node characteristics and edge characteristics, the statistical relationships (correlation or else) between these values need to be provided; overall and within the different cell types. Certain parameters are likely to be trivially linked, but this may report important aspects of the cellular networks organizations that are currently ignored in this version of the manuscript. In particular, which are the local (cell parameters) that influence BC-is a cell able to control in some ways its own BC?

3) The Discussion is focused narrowly, and mostly on the authors own work. The Discussion would be more valuable if it were expanded to consider: a) other plant tissues and other work in the field of plant tissue patterning, b) animal cell organization (embryo patterning in *C. elegans, D. melanogaster*, Neurons…) where cell migration is operating and how the authors see cell networks in this dynamical context.

[Editors' note: further revisions were requested prior to acceptance, as described below.]

Thank you for resubmitting your work entitled "Topological analysis of multicellular complexity in the plant hypocotyl" for further consideration at *eLife*. Your revised article has been favorably evaluated. Improvements on the clarity of the writing, statistical measurements and inclusion of functional data on atrichoblast vs. trichoblast files have made this version much stronger.

We intend to recommend this manuscript be accepted, but a few changes to the explanation of the fluorescein experiment (in Figure 3) would be helpful. The names of atrichoblast and trichoblast cell files (while not incorrect) are somewhat confusing because these files, in the hypocotyl, do not produce hairs. Experiments involving uptake of substances from the media will immediately make people think about hair and non-hair cells of the root, and it is natural to begin to think that. Rewording or putting in an additional sentence to explain that the difference in movement in the cells you are monitoring is *not* due to morphological differences in these cells themselves (and that they are some distance from the site of uptake) would eliminate this confusion.

---

## [Author Response]

*Major issues:*

*1) eLife papers describing new techniques or analysis approaches like this one should be accessible and interesting to a wide audience. Several reviewers felt that the authors did not yet make a persuasive case that they are discovering general principles here, and for this work to be useful for the developmental biology community, the connection between the detailed measurements and biological problems must be made very clearly. This "Major issues" section enumerates what they considered to be necessary changes to be made in the revision:*

*A) The connection between what is measurable and what can be interpreted as the biological significance of the measurement needs to be done with some care. For example, the authors interpret BC as meaning that "atrichoblast cell files are topologically poised to mediate the optimized movement of information" and "having the capacity to control information movement through a limited number of local interacting partners". This is a leap. For example, BC may highlight nodes that form the sole pathway between two otherwise disconnected parts of the network. In that case the argument that BC signifies the potential to control flow (of information, or anything else) is justified. However, BC can be high in many other contexts. In the cellular contact network in particular there will be many ways that information can pass between cells, as this network resembles a lattice. A high BC node therefore could lie on a slightly shorter path, but there will be many other paths, perhaps even of similar length. The equivalence between high BC and a capacity to control is therefore much less obvious. The authors should put less emphasis on this interpretation of BC. They may still have a point here, but at present the manuscript is overstating the connection between BC and control. This is linked to a general feeling that the equivalence of cellular networks and networks of directed information flow (based on the evidence presented in this paper), is overstated.*

This point is well taken considering the lattice-like structure of these spatially embedded networks. While experimental evidence demonstrating the preferential directional movement of small molecules along shorter paths within the hypocotyl epidermis has been provided in Figure 3, the nature of these optimized paths in the control of this movement is less clear. Throughout the text we have de-emphasized the role of BC in controlling molecular movement across these networks to address this comment.

*B) Several reviewers felt there was not a clear path between the technique described here and genuine insight into developmental biology. The clearest way to show that this technique can lead to new biological insights is to provide such an insight. This may be achieved by revisiting old experiments in the literature that had some contradictory or hard to understand elements and showing how the approach presented in this paper can explain the discrepancy/discordance in the data. Alternatively, an observation made here should be predictive of a behavior; for example, the authors describe a difference in connectivity between atrichoblasts or trichoblasts: what would you predict are different properties of a hormone or Ca or physical signal sent through one or the other of these cell types? And how would you confirm this experimentally?*

To more clearly link the topological analysis of global cellular connectivity and insight into developmental processes within the plant hypocotyl, we have performed two major changes to the manuscript.

The first is modifications to the figures and text to principally focus upon the analysis of the higher-order properties of epidermal cell patterning. The targeted analysis of this subset of cells within the hypocotyl describes the emergence of reduced path length specifically within the atrichoblast cell type in *Arabidopsis*, and plasticity in the emergence of this property across different genetic backgrounds. This provides a clear message as to how a higher order principle of global cellular configurations can emerge in different genetic contexts.

The second major modification to the manuscript is the functional exploration of these contrasting path lengths within different epidermal cell types. A series of experiments examining the movement of fluorescent molecules across *Arabidopsis* hypocotyls was performed. Seedlings from the 3 ecotypes topologically analysed in this work were transferred to media containing the small fluorescent molecule fluorescein. Following an incubation period, the seedlings were imaged using confocal microscopy to quantify the abundance of fluorescein within the different cell types of the epidermis using quantitative 3D image analysis. In each Col and L*er*, where atrichoblasts lie upon shorter paths, fluorescein was significantly more concentrated in the non-hair cell type (Figure 3). In Cvi where no difference in path length between epidermal cells is observed, fluorescein did not show a significant difference in its abundance between epidermal cells. Reduced path lengths can therefore provide optimized conduits for the movement of molecules across *Arabidopsis* hypocotyls. The topological proxy of BC can predict this long-range movement at single cell resolution in these complex organs.

We proposed this higher-order property of epidermal cell organization may bridge a structure-function relationship within epidermal cell patterning (subsection “Quantification of cell type-specific molecular movement through *Arabidopsis* hypocotyls”, fourth paragraph). The presence of a division of labour between nutrient-acquiring root hair cells and their adjacent non-hair cells may be present, such that newly acquired molecules are transferred from trichoblast to atrichoblast cells. By moving solutes out of root hairs, these cells can maintain low intracellular concentrations facilitating further nutrient uptake from the soil. Their adjacent atrichoblast cells receive these molecules, and are topologically optimized for long-range longitudinal movement up the plant axis. This organizational configuration between these two cell types provides a mechanism for the optimized uptake and transport from the root up through the hypocotyl. Plasticity in the presence of this property suggests this feature may play an adaptive role in certain contexts but not others, and is discussed further below.

*C) The rationale explaining how the authors picked the different plant species and the different mutants are not crystal clear. For the flow of the ideas and the understanding of the reader, it would be nice to include two paragraphs explaining the general concept and why the authors chose these plant species (first paragraph) and mutants (second paragraph).*

We have added paragraphs describing the rationale for the selection of the species in this study in the subsection “Comparison of topological features of diverse plant species” and the selection of mutants in the second paragraph of the subsection “Topological analysis of cellular patterning mutants”.

*D) There needs to be a more coherent section at the end summarizing precisely what principles hold across ecotypes, species, and mutants (backed up by statistical significance measures). A more rigorous set of conclusions of this kind would provide an adequate justification for the rather strong and general claims made in the abstract. The question that needs to be clearly answered in this summary is: What can network measures predict about plant cells?*

The question – “What can network measures predict about plant cells?” – is addressed in newly added Figure 3 and 8. In Figure 3 we demonstrate that path length, as determined using BC, can predict the movement of molecules across the hypocotyl epidermis at single cell resolution. This ability to understand intercellular communication on an organ level provides a developmentally relevant context for the topological measures used. The presence of plasticity in atrichoblast path length and the genetic factors that contribute towards the formation of this emergent property provide further insight into this important feature.

Figure 8 explores the topological principles across genotypes, species and mutants used in this study, describing the overarching principles of hypocotyl design at cell type-specific resolution. This revealed that the topological properties of each epidermal cell type are distinct from all other cell types, while the outer cortical cells share the greatest similarity with the endodermis. Unexpectedly, the inner cortex is topologically distinct from all other cell types. These principles hold true for all wild-type genotypes with the exception of poppy, which has a distinct cellular topology from other samples examined. This basal eudicot is the most evolutionarily ancient species examined and also has the longest path lengths of all in this study. The emergence of shorter paths in the hypocotyl may therefore represent a novel evolutionary innovation (subsection “Comparison of topological features of diverse plant species”, last paragraph). The developmental significance of this higher-order property is highlighted by the demonstration that molecules and information preferentially follow these conduits altering the chemical composition and events, which occur across organs.

*E) There are some technical issues in the way data are presented or interpreted. For example, weighted BC (subsection “Topological analysis of the wild-type Colombia Arabidopsis hypocotyl”, tenth paragraph). Generalisations of BC to weighted networks are not entirely straightforward, as there is more than one way to interpret path length in weighted networks. One way for example is to use the reciprocal of the weight as the contribution that a link makes to the overall path length. The authors should clarify what version of weighted BC they are using. It seems that the inverse of the connecting cell wall area was used as the weight itself, but weighted path length calculations often use the reciprocal of the weight, which means that there might be a double reciprocal here? In any case the definition of the weight and the exact algorithm should be made explicit.*

The double reciprocal was not used, and this has been clarified in the main text as: “In this case, weighted betweenness centrality was calculated using the reciprocal of the connecting cell wall area (edge weight), so that large cell-cell interactions generate small weights, and lead to the calculation of shorter paths (Newman, 2010).”

*2) In general, arguments throughout the manuscript rely a lot on visual comparisons of distributions, which are not always convincing. There are plenty of rigorous statistical tools for comparing two distributions, which would return a measure of significance for the difference between the distributions. Calculating such significance values for all the relevant comparisons of distributions in the paper would make the authors' arguments much stronger (providing the statistics support their claims). In addition, since you are able to study many different parameters extracted from the plant tissues (e.g. cell characteristics, node characteristics and edge characteristics, the statistical relationships (correlation or else) between these values need to be provided; overall and within the different cell types. Certain parameters are likely to be trivially linked, but this may report important aspects of the cellular networks organizations that are currently ignored in this version of the manuscript. In particular, which are the local (cell parameters) that influence BC-is a cell able to control in some ways its own BC?*

We thank the reviewers for this suggestion to perform statistical tests on all data presented as this has increased the strength with which statements have been made in this work, and clarified their interpretation.

For all distributions presented in both main and supplemental figures, we performed Kolmogorov–Smirnov significance tests. This establishes whether two distributions are significantly different from one another, and an asterisk is added in each figure to indicate where significant differences are present. Changes in the text throughout the manuscript accompany these significant tests.

The question – “which are the local (cell parameters) that influence BC-is a cell able to control in some ways its own BC?” – is explored through the addition of a new figure (Figure 6). Here, mutual information using Shannon entropy enables the extent to which one variable is related to another (subsection “The relationship between geometric, local and global topological features of cells and cell networks”). This analysis demonstrated that the size of a cell has little to no bearing as to the length of the path that it lies upon. Similarly, the immediate number of neighbours of a cell and its path length are only weakly associated. Path length is therefore a property emerging from the global context within which a cell is situated in an organ, and is not associated with either cell geometry or local topology.

*3) The Discussion is focused narrowly, and mostly on the authors own work. The Discussion would be more valuable if it were expanded to consider: a) other plant tissues and other work in the field of plant tissue patterning, b) animal cell organization (embryo patterning in C. elegans, D. melanogaster, Neurons…) where cell migration is operating and how the authors see cell networks in this dynamical context.*

A much broader discussion covering the topics suggested and others, in addition to a wide range of new references, has been added to the start of the Discussion section. This text describes broad aspects of cellular patterning research across both plant and animal systems. Included is also a discussion of dynamic cellular topologies observed in animal systems and a proposed mathematical framework to study these.

[Editors' note: further revisions were requested prior to acceptance, as described below.]

*We intend to recommend this manuscript be accepted, but a few changes to the explanation of the fluorescein experiment (in Figure 3) would be helpful. The names of atrichoblast and trichoblast cell files (while not incorrect) are somewhat confusing because these files, in the hypocotyl, do not produce hairs. Experiments involving uptake of substances from the media will immediately make people think about hair and non-hair cells of the root, and it is natural to begin to think that. Rewording or putting in an additional sentence to explain that the difference in movement in the cells you are monitoring is not due to morphological differences in these cells themselves (and that they are some distance from the site of uptake) would eliminate this confusion.*

To address your suggestion, we have added a couple of lines of explanatory text in relation to Figure 3 to help clarify the interpretation of these experimental data and the role of epidermal cell types in the hypocotyl: “Epidermal cells in the hypocotyl typically do not produce root hairs, despite having each trichoblast and atrichoblast cell identities. The observation of the passage of fluorescein along the hypocotyl occurs following molecular uptake in roots, where hairs are produced, and represents long distance transport along the length of the plant rather than local uptake.” We hope this clarifies any confusion on the part of the reader, and are happy to modify accordingly if not.